# Detection and Genetic Characterization of Hepatitis B and D Viruses: A Multi-Site Cross-Sectional Study of People Who Use Illicit Drugs in the Amazon Region

**DOI:** 10.3390/v13071380

**Published:** 2021-07-15

**Authors:** Ronylson José S. Silva, Raquel Silva do Nascimento, José Augusto J. Oliveira-Neto, Fabricio Quaresma Silva, Juliana Nádia F. Piauiense, Camila Moraes Gomes, Luiz Marcelo L. Pinheiro, Rafael Lima Resque, João Renato R. Pinho, Emil Kupek, Benedikt Fischer, Luiz Fernando A. Machado, Luísa Caricio Martins, José Alexandre R. Lemos, Aldemir B. Oliveira-Filho

**Affiliations:** 1Programa de Pós-Graduação em Biologia Ambiental, Universidade Federal do Pará, Bragança 68600-000, PA, Brazil; ronylson100@gmail.com (R.J.S.S.); siilvaraquel@gmail.com (R.S.d.N.); milagomes01@hotmail.com (C.M.G.); 2Instituto de Estudos Costeiros, Universidade Federal do Pará, Bragança 68600-000, PA, Brazil; augustoliveiraneto@gmail.com (J.A.J.O.-N.); fabioquaresmabio@hotmail.com (F.Q.S.); 3Programa de Pós-Graduação em Saúde na Amazônia, Universidade Federal do Pará, Belém 66055-240, PA, Brazil; jnfpi@hotmail.com (J.N.F.P.); caricio@ufpa.br (L.C.M.); 4Faculdade de Ciências Biológicas, Campus do Marajó, Universidade Federal do Pará, Soure 68870-000, PA, Brazil; lmpinheiro@ufpa.br; 5Departamento de Ciências Biológicas e da Saúde, Universidade Federal do Amapá, Macapá 68903-419, AP, Brazil; rafaelresque@gmail.com; 6Instituto de Medicina Tropical, Universidade de São Paulo, São Paulo 05403-000, SP, Brazil; jrrpinho@usp.br; 7Departamento de Saúde Pública, Universidade Federal de Santa Catarina, Florianópolis 88040-900, SC, Brazil; emil.kupek@ufsc.br; 8Centre for Applied Research in Mental Health and Addiction, Faculty of Health Sciences, Simon Fraser University, Vancouver, BC V6B 5K3, Canada; benedikt_fischer@sfu.ca; 9Faculty of Medical and Health Sciences, University of Auckland, Auckland 1023, New Zealand; 10Departamento de Psiquiatria, Universidade Federal de São Paulo, São Paulo 04038-000, SP, Brazil; 11Instituto de Ciências Biológicas, Universidade Federal do Pará, Belém 66075-110, PA, Brazil; lfam@ufpa.br (L.F.A.M.); jalemos@ufpa.br (J.A.R.L.); 12Núcleo de Medicina Tropical, Universidade Federal do Pará, Belém 66055-240, PA, Brazil

**Keywords:** epidemiology, HBV, HDV, exposure, genotypes, mutations, interventions, Amazon

## Abstract

Hepatitis B (HBV) and delta (HDV) viruses are endemic in the Amazon region, but vaccine coverage against HBV is still limited. People who use illicit drugs (PWUDs) represent a high-risk group due to common risk behavior and socioeconomic factors that facilitate the acquisition and transmission of pathogens. The present study assessed the presence of HBV and HBV-HDV co-infection, identified viral sub-genotypes, and verified the occurrence of mutations in coding regions for HBsAg and part of the polymerase in HBV-infected PWUDs in municipalities of the Brazilian states of Amapá and Pará, in the Amazon region. In total, 1074 PWUDs provided blood samples and personal data in 30 municipalities of the Brazilian Amazon. HBV and HDV were detected by enzyme-linked immunosorbent assay and polymerase chain reaction. Viral genotypes were identified by nucleotide sequencing followed by phylogenetic analysis, whereas viral mutations were analyzed by specialized software. High rates of serological (32.2%) and molecular (7.2%) markers for HBV were detected, including cases of occult HBV infection (2.5%). Sub-genotypes A1, A2, D4, and F2a were most frequently found. Escape mutations due to vaccine and antiviral resistance were identified. Among PWUDs with HBV DNA, serological (19.5%) and molecular (11.7%) HDV markers were detected, such as HDV genotypes 1 and 3. These are worrying findings, presenting clear implications for urgent prevention and treatment needs for the carriers of these viruses.

## 1. Introduction

Hepatitis B virus (HBV) infection is a global public health problem with approximately two billion people worldwide infected in the past and 257 million chronic carriers [1]. HBV-related progression of liver disease to severe forms such as liver cancer entails a heavy burden for many countries [1,2], despite the availability of safe and effective vaccines [3,4].

Taxonomically, HBV belongs to the Hepadnaviridae family and possesses a partial double-strand DNA genome of approximately 3.2kb. Ten HBV genotypes (A–J) have already been described, with nucleotide sequence divergences greater than 8% in the entire viral genome [5]. Most genotypes have several subgenotypes that may lead to different outcomes in chronicity after infection, disease progression, and responses to IFNα treatment [5,6]. Genotypes A, D, and F are commonly detected in Brazilians with HBV [7,8]. HBV mutations in the regions of the polymerase (P), core (C), surface antigen (S), and X proteins are considered important because they can influence the outcome of the HBV infection, increasing the risk of developing liver cirrhosis and hepatocellular carcinoma in response to treatment with antiviral drugs and in the immunological escape of the vaccine [9,10,11,12].

The World Health Organization (WHO) classifies hepatitis B endemicity according to the prevalence of its serological marker known as the hepatitis B surface antigen (HBsAg): low (<2%), intermediate–low (2–4%), intermediate–high (5–7%), and high (>8%) [1]. The Brazilian Amazon (northern Brazil) is characterized as a region of high HBV endemicity, the largest one among Brazilian regions. However, this endemicity is not uniform [13,14,15,16,17]. In recent decades, the prevalence of HBsAg carriers has fluctuated from 0% to 20% in riverside communities and indigenous tribes in the Brazilian Amazon, and in a smaller proportion in urban populations, from 0 to 6% of HBsAg carriers [18,19,20,21]. Occurrences of HBV-associated diseases and their sequelae have been recorded more frequently in the states of Acre, Amazonas, and Rondônia than in other Brazilian states in the Amazon region [20,22]. For example, a very high lethality has been associated with the hepatitis delta virus (HDV) infection in HBsAg carriers in the state of Amazonas [22].

HDV is also endemic in the Amazon region, especially in the Brazilian states of Acre and Amazonas [23,24,25]. Among chronic HBV carriers, it is estimated that about 15–20 million were also infected with HDV [26]. HDV depends on HBV infection to complete its life cycle, as HDV requires protein S from HBsAg for its assembly and propagation [27]. HDV has a circular single-stranded RNA genome of approximately 1.7 kb [28]. Based on phylogenetic analyses, it has been classified into eight genotypes (1–8) [27]. HDV genotypes 1, 3, and 8 are found in South America, with genotype 3 considered to be original and prevalent in the Amazon region [29,30,31,32,33]. HDV infection is often associated with severe liver disease [32,34].

HBV is transmitted by the sexual and parenteral routes, and HDV is transmitted by the parenteral route and with evidence of sexual transmission. People with high-risk sexual activity are at increased risk of HBV and HDV infection. The transmission of these two viruses is also common due to inadequate screening before surgical procedures and blood transfusion in endemic areas [2,4,27,35]. People who use illicit drugs (PWUDs) are an important group vulnerable to viral infections as a result of socioeconomic factors and risky behaviors that facilitate viral acquisition and transmission, such as sharing instruments for drug use, long time and high frequency of drug use, the type of illicit drug used, unprotected sex, multiple sexual partners, and exchange of sex for money/drugs [13,35,36,37,38]. In the Amazon region, high rates of exposure to HBV (22.7%–36.7%) and genotypes A, D, and F were identified using real-time PCR in studies conducted with PWUDs [13,37,39]. To date, the presence of HDV in PWUDs has not yet been reported. In addition, the presence of occult HBV infection (OBI) has been recorded in the Amazon region, including in PWUDs [13,37,40,41,42]. OBI is defined as HBsAg-negative and anti-HBc positive/negative with HBV DNA detectable in serum and liver tissue. Most cases of OBI are asymptomatic and can be found in one of the following scenarios: (i) a window period of acute HBV infection, (ii) detectable HBV DNA and undetectable HBsAg in patient serum without a previous history of overt HBV infection, and (iii) patients with a history of chronic HBV infection [43].

Vaccination coverage against HBV is still precarious in the Amazon region, although it has been considered an endemic area for decades [20,44]. Given the need to understand the epidemiological scenario of HBV infection and HBV-HDV co-infection among high-risk groups in the largest Brazilian region, the present study assessed the presence of HBV and HBV-HDV, identified viral genotypes, and verified the occurrence of mutations in coding regions for HBsAg and part of the polymerase in HBV-infected PWUDs in the municipalities of the Brazilian states of Amapá and Pará, in the Amazon region, as well as indicated key interventions for the control and prevention of these viral infections.

## 2. Materials and Methods

### 2.1. Study Design and Data Collection

This cross-sectional study was based on biological and self-reported socio-behavioral data from a respondent-driven sample of 1074 PWUDs from epidemiological studies conducted in municipalities located in the Brazilian states of Amapá and Pará, in the Amazon region (Figure 1) [13,37,39]. All subjects were community-recruited by the snowball technique [45]. Study eligibility criteria were (1) use of illicit drugs in the last three months, (2) 18 years of age or older, (3) not impaired from drug use during the study assessment, and (4) willing and consenting to complete the study protocol, i.e., to provide a biological sample and to complete the epidemiological assessment. All samples and data were collected between March 2013 and December 2018.

### 2.2. Laboratory Tests

All samples of PWUDs were evaluated for the presence of different laboratory markers for hepatotropic viruses. According to the manufacturers’ instructions, this study evaluated the presence of serological markers for HBV: HBsAg (Murex HBsAg Version 3, DiaSorin; sensitivity 100.0% and specificity 99.9%), total anti-HBc antibodies (Murex anti-HBc (total), DiaSorin; sensitivity 100.0% and specificity 99.7%), and anti-HBs (Monolisa Anti-HBs PLUS, Bio-Rad; sensitivity 99.2% and specificity 99.4%). All PWUDs samples were also subjected to viral DNA isolation (QIAamp DNA Blood Mini Kit; Qiagen, Hilden, Germany) and later submitted to amplification using real-time polymerase chain reaction (PCR) to evaluate the presence of HBV DNA, with detection limit of 50 IU/mL (2.8 × 10^2^ HBV copies/mL) [46].

All samples with positive results for the presence of the HBV DNA were evaluated for the presence of total antibodies to hepatitis delta antigen (anti-HD) (ETI-AB-DELTAK-2, Bio-Rad; sensitivity 99.4% and specificity 99.0%) by enzyme immunoassay (EIA), and they were also submitted to viral RNA isolation (QIAmp Viral RNA Mini extraction kit, Qiagen, Hilden, Germany), transcription from RNA to complementary DNA (cDNA) (High-Capacity cDNA Reverse Transcription kit; Applied Biosystems, Foster City, CA, USA), and, subsequently, evaluation of the presence of HDV cDNA (i.e., HDV RNA) using real-time PCR, with detection limit of 7.5 × 10^2^ HDV copies/mL, as described previously [47]. In all reactions, an internal control (TaqMan Exogenous Internal Positive Control Reagents kit; Applied Biosystems, Foster City, CA, USA) was used to assess the presence of inhibitors in the reaction, thus reducing the possibility of false-negative results. All cases of exposure to HBV (including cases of OBI) and HBV-HDV were tested twice to confirm the occurrence of the events.

### 2.3. Amplification and Sequencing of Genomic Fragments

All samples of HBV DNA were submitted to Nested-PCR for fragment amplification with approximately 1300 base pairs (bp), comprising the whole HBsAg and part of the polymerase coding genes [48]. The samples with HDV RNA were submitted to nested-PCR for fragment amplification of approximately 320 bp, comprising part of the delta antigen genomic region [49]. PCR products from the second-round reaction (nested PCR) were purified using the Charge Switch PCR Clean-Up Kit (Invitrogen, Waltham, MA, USA) and were then sequenced using the BigDye Terminator v3.1 Cycle Sequencing Kit (Applied Biosystems, Foster City, CA, USA). Each amplicon was analyzed in both sense and antisense directions. Sequencing of nucleotides was performed in ABI 3130 Genetic Analyzer (Applied Biosystems, Foster City, CA, USA).

### 2.4. Phylogenetic Analysis

In this study, nucleotide sequences were aligned in ClustalW [50] implemented in the BioEdit software [51]. Nucleotide sequences of references from the HBV and HDV genotypes/subgenotypes, accessed at the National Center for Biotechnology Information (https://www.ncbi.nlm.nih.gov/pubmed, accessed on 1 April 2021), were added to the referred viral sequence alignments. The jModeltest2 software [52] was used to indicate the replacement model to be used in the construction of the phylogenetic trees of HBV and HDV, according to the Akaike information criterion (AIC) and Bayesian information criterion (BIC). Maximum likelihood and Bayesian inference trees were made in RaxML [53] and MrBayes 3.2.1 [54], respectively. The trees obtained were edited using the software FigTree v1.4 [55]. The Geno2pheno-HBV program was also used to indicate the genotypes of HBV [56].

### 2.5. Mutation Identification

The HBV nucleotide sequences obtained in this study were examined for vaccine escape mutations (Gene S HBsAg) and drug resistance mutations (Gene P RT Region) using the Geno2pheno-HBV programs [56] and HIV-Grade: HBV-Tool [57] in their respective reference banks. Finally, BioEdit software was used for visual confirmation of HBV mutations [51]. The HBV nucleotide sequences obtained in this study were translated into amino acids, aligned, and compared with HBV consensus sequences generated for each identified genotype/subgenotype.

### 2.6. Statistical Analysis

All study data were entered into an Excel database (Microsoft Corporation) and converted to SPSS (IBM). Absolute (N) and relative (%) frequencies of the variables were used for description; 95% confidence intervals were calculated for the prevalence of viral genotypes and subgenotypes. Statistical analyses were conducted using the SPSS 23.0 software for Windows.

### 2.7. Ethical Review

This study was approved by the Committee for Ethics in Research of the Núcleo de Medicina Tropical of the UFPA in Belém, Brazil (CAAE: 37536314.4.0000.5172). The subjects provided their written informed consent to participate in this study.

## 3. Results

### 3.1. Sample Characteristics

In total (*n* = 1074), 766 PWUDs were accessed in 19 municipalities in the state of Pará and additional 308 PWUDs in 11 municipalities in the state of Amapá (Figure 1). The average number of PWUDs in each municipality was 36 (standard deviation = 35). The highest and lowest number of PWUDs were obtained in Breves (*n* = 180) and Tartarugalzinho (*n* = 15), respectively.

In this study, most of the PWUDs were male, young (18–29 years old), black or mixed race, heterosexual, unmarried, with low education, and low monthly income. Regarding the use of drugs and risky behaviors, many subjects used crack-cocaine (or oxi), used drugs daily, shared equipment for drug use, had unprotected sex, and had more than 10 sexual partners. Injecting drugs and trading sex for money or drugs were additional characteristics relevant to viral spread (Table 1).

### 3.2. Laboratory Diagnosis of Viruses

Overall, 346 (32.2%) PWUDs showed some serological marker for HBV, of which 50 (4.7%) had positive results for HBsAg antigen, 272 (25.3%) had anti-HBc antibodies with or without anti-HBs antibodies, and 24 (2.2%) showed only anti-HBs antibodies, which indicated that they had been vaccinated and were not exposed to HBV (Table 2). In addition, 77 (7.2%) PWUDs had detectable HBV DNA. Viral load ranged from 5.6 × 10^2^ to 3.1 × 10^4^ HBV copies/mL. All PWUDs with positive results for HBsAg (*n* = 50) also had positive results for HBV DNA (viral load: 1.2 × 10^4^ and 3.1 × 10^4^ HBV copies/mL) (Table 2). Among the 1024 PWUDs with negative results for HBsAg, 5 PWUDs with positive results only for the anti-HBc marker and 11 subjects with positive results for both anti-HBc and anti-HBs markers were also found to have HBV DNA (viral load: 5.6 × 10^2^ to 4.8 × 10^3^ HBV copies/mL). In this study, 11 subjects had DNA-HBV and had no serological marker for HBV (viral load: 5.2 × 10^3^ to 7.6 × 10^3^ HBV copies/mL). In total, 27 (2.5%) cases of OBI were detected (viral load: 4.4 × 10^3^ to 9.6 × 10^3^ HBV copies/mL). 

Among PWUDs with HBV DNA (50 with positive results for HBsAg, and 27 with negative results for HBsAg), 15 (19.5%) had anti-HDV antibodies and 9 (11.7%) had HDV RNA (viral load: 8.4 × 10^2^ to 6.8 × 10^3^ HDV copies/mL) (Table 3). All PWUDs with positive results for HDV also had tested positive for HBsAg. Until participating in this study, none of the PWUDs had done any laboratory tests for HBV and/or HDV-HBV. All study subjects said they did not undergo antiviral treatment for viral hepatitis.

### 3.3. Genotypes and Subgenotypes

HBV genotypes A (53.2%), D (29.9%), and F (16.9%) were identified among PWUDs. The subgenotypes A1, A2, D4, and F2a had the highest frequencies (Table 2). Among 27 PWUDs with OBI, subgenotypes A1 (55.6%), A2 (7.4%), D2 (3.7%), D3 (7.4%), D4 (18.5%), F1a (3.7%), and F1b (3.7%) were detected. Genotype 3 predominated among PWUDs with HDV RNA (Table 3). Among PWUDs infected with HBV and HDV, the majority had HDV genotype 3 and HBV subgenotype A1 or A2 (Table 4). The phylogenetic trees of HBV (Appendix A) and HDV (Appendix A) among PWUDs in the Amazon region, as well as information needed for their construction (Appendix A), can be accessed in the Appendix A.

### 3.4. Identification of HBV Mutations

In 77 nucleotide sequences, three vaccine escape mutations (G130R, P120L, and A128V) and one antiviral resistance mutation (M204I) were identified (Table 5). Two vaccine escape mutations (P120L and G130R) were detected in PWUDs with OBI. The analyses made in the Geno2pheno-HBV [55] and HIV-Grade: HBV-Tool [56] programs indicated the same results, and the four mutations described were visualized in the alignment of amino acid sequences using BioEdit software [50].

## 4. Discussion

The present study identified relevant information on the epidemiology of HBV infection and HBV-HDV co-infection among PWUDs in the Brazilian states of Amapá and Pará, in the Amazon region. This can be used to direct actions for diagnosis and treatment of current viral infections and to prevent new infections, as well as serving as a strong incentive to promote the health of PWUDs in this Brazilian region and other parts of the world with similar characteristics.

The first important finding was that 30.0% of PWUDs were exposed to HBV, as indicated by the presence of either HBsAg or anti-HBc or both. This percentage is higher than the values reported for most PWUDs from two other Brazilian regions, such as Salvador (0.0–6.2%) in the Northeast, and Campo Grande, Cuiabá, and Goiânia (7%–14.0%) in the Midwest [44,58,59,60]. Other countries reported lower exposure to the HBV: France (1.4%), China (13.7%–25.5%), the United States (25.6%), and Luxembourg (29.1%) [61,62,63,64]. This finding confirms the vulnerability of PWUDs to HBV exposure in the Amazon region. The same goes for other viruses, such as HCV, HIV, and HTLV, whose prevalence was associated with age (>35 years), sex (male), longer use of illicit drugs, factors related to parenteral transmission (use of injectable drugs and shared use of equipment for drug use), risky sexual behavior (unprotected sex, multiple sexual partners, sex with other PWUDs), and socioeconomic marginalization (low education, low monthly income, lack of access to health services) [36,37,45,65].

Second, this study evaluated the presence of HBV DNA in all samples, which allowed us to detect OBI. The prevalence of HBV DNA and OBI among PWUDs was 7.2% and 2.5%, respectively. This OBI rate is similar among people who use non-injectable drugs in the Midwest of Brazil (2.7%) [44]. In a geographic region endemic for hepatitis B and with limited screening programs for the diagnosis of HBV, such as Amazonia, OBI can represent a risk for transmission of the virus through blood donation, organ transplants, hemodialysis, and other health procedures. Additionally, OBI contributes to maintaining the circulation of HBV among PWUDs. Besides, OBI can be transmitted to the general population when screening for HBV infections does not use PCR, i.e., based only on serological tests or rapid qualitative tests, as is done in most cities in the Brazilian Amazon. In the present study, there was no clinical follow-up and no more detailed investigation of the subjects’ life histories, so it was not possible to accurately identify the clinical profile of PWUDs. OBI has already been reported in the Amazon region, thus reinforcing the need for molecular screening of HBV [42], particularly in high-risk groups such as PWUDs.

Third, the frequency of HBV genotypes and subgenotypes among PWUDs in the Amazon region identified HBV genotype A with the highest frequency, followed by genotypes D and F. This pattern is consistent with other studies conducted in the Brazilian Amazon, both in the general population and in high-risk groups (PWUDs and female sex workers) [7,8,13,37,38,39,66]. Among people with OBI, the presence of genotypes A (A1 and A2) and F (F2) has been recorded in the Amazon region [42]. However, this study detected a greater number of HBV subgenotypes, such as A1, A2, D2, D3, D4, F1a, and F1b. These findings can be a result of the miscegenation of Amerindians, Africans, and Europeans due to migration. The interaction of people with risk factors (low education, low monthly income, limited public health service, and several socioeconomic problems) and risky behaviors (unprotected sex, injecting drug use, sharing equipment for drug use, exchange of sex for money/drugs) may have contributed to the variety of HBV genotypes in this Brazilian region. 

The evidence of this process can be seen through the frequency and distribution of HBV subgenotypes. A1, D3, and D4 had the highest frequencies, and a diversity of other subgenotypes recorded in Brazil and in other South American and Central American countries were detected in PWUDs. The high frequency of the A1 subgenotype is consistent with epidemiological studies conducted in Brazil [41,42,67,68,69,70]. Subgenotypes D4, D3, and D2 were identified in PWUDs and have already been detected in people from Brazilian states in the Amazon region, such as Amazonas (D3 and D4), Rondônia, and Roraima (D2, D3, D4), but with different frequencies [29,69,70]. F1a and F2b were detected in Central America (Costa Rica and El Salvador) and Venezuela, respectively [71,72]. This is the first record of these F subgenotypes in Brazil. F1b has already been detected in Alaska, Argentina, Brazil, Chile, Colombia, Peru, and Venezuela. In Brazil, F1b was reported in the northeastern and southern states, and more recently for the first time in the Brazilian Amazon [71,72,73]. Subgenotype F2a has been restricted to Venezuela and Brazil, with high frequency in the Brazilian Amazon [71,72,74]. F3 is commonly found in Panama, Colombia, and Venezuela, being detected for the first time in the Brazilian Amazon in people who lived in the state of Roraima [69,71]. The risk behaviors of PWUDs allow them to acquire and transmit a variety of HBV subgenotypes circulating in the Amazon region, both in Brazilian states and in other American countries, such as Colombia, Peru, and Venezuela, as shown in the first report on HBV subgenotypes F1a and F2b in Brazil.

Fourth, the identification of antiviral resistance and vaccine escape mutations were detected in HBV-infected PWUDs. Mutations in the HBV genome, such as vaccine escape (gene S) and resistance to antivirals (gene P), pose a public health challenge [75,76]. Escape vaccine mutations are associated with changes in the determinant ‘‘a’’ of HBsAg (aa 124 to 147), which can affect HBsAg antigenicity [75,77]. This is dangerous as it makes it difficult to recognize HBsAg and can result in the vaccine-induced escape of the host immune response [78,79,80,81]. Vaccine escape mutations detected in the present study are not commonly described in their places of origin, as is the case of G130R, site G130. P120T is more commonly described at position P120 than P120L observed in our results [82]. Interestingly, another record was made in Rio de Janeiro, where P120L was identified in a person with OBI and genotype D2 [81]. A128V mutation, A128 site, has been reported in India and in cases of OBI and genotype D2 [83].

Clinical protocols for therapeutic guidelines for chronic hepatitis B have been offered to Brazilians, free of charge, for two decades. Prolonged use of nucleotide analogs, such as lamivudine, adefovir dipivoxil, and entecavir, can cause the emergence of drug-resistant strains [83]. Although no PWUD was treated for chronic hepatitis B, one antiviral resistance mutation was detected. The rtM204I is a primary resistance mutation, located in the C domain of the YMDD region of the P gene, found more easily individually, different from the mutation in the same site (rtM204V) with valine replacing the initial amino acid, methionine, which has been associated with resistance to lamivudine, telbivudine, and entecavir [10,84,85]. Some studies suggest that this mutation occurs more frequently in HBV genotypes B, C, or D [86,87,88]. However, similar to this study, another report also detected the occurrence of rtM204I in genotype A (subgenotype A1) of HBV [12]. In general, the occurrence of various HBV mutant strains underlines the need for careful monitoring, especially in a high-risk and difficult-to-access group such as PWUDs.

Fifth, this is the first report of HBV-HDV co-infection among PWUDs in South America. High exposure to HDV in this vulnerable group was expected and is consistent with the HDV endemic status of the Amazon region [25,89,90,91]. The prevalence of anti-HDV antibodies among PWUDs was still lower than that found in riverside communities in the states of Acre and Amazona (about 66%) [25], but it was higher than in the Midwest (2.5%), Southeast (1.7%), and Northeast (0.8%) of Brazil [90]. The prevalence of anti-HDV antibodies in people who use injectable drugs in several countries (Thailand (21.8%), Poland (23.2 %), Iran (28.6%), Greece (35.0%) and Australia (40.9%)) was also higher than the value registered here [92]. This is possibly a reflection of the increased transmissibility of HDV-HBV among people who use injectable drugs compared to people who use non-injectable drugs. HDV genotypes 1 and 3 were detected in PWUDs. The predominance of genotype 3 is consistent with the findings of epidemiological studies in South America, and the identification of HDV genotype 1 reinforces the perspective of its worldwide distribution [30,31,32]. The co-infection of HDV genotype 3 and HBV genotype A (A1 or A2) observed in most PWUDs is also consistent with the studies of people residing in the Brazilian states of Acre, Amazonas, and Pará, and it reinforces the regional pattern of this co-infection [49,93,94]. To date, there is little evidence of the implications of the association of genotypes A and 3 on the clinical course of co-infected individuals, only a report of a lower viral load [95]. Interestingly, genotypes F and 3 have already been associated with one of the serious forms of this co-infection, fulminant hepatitis [95,96,97]. In this study, three PWUDs with HBV-HDV were diagnosed with genotypes F and 3, but there was no clinical evaluation of these participants.

All these findings point to several important implications for interventions in the Amazon region. As a main preventive measure, HBV vaccination should be extended to high-risk groups. Although an HBV vaccination program has been developed in the Amazon region in recent decades [14,20], epidemiological studies have indicated very low vaccine coverage in PWUDs and FSWs in this area, thus indicating the urgent need to vaccinate members of these vulnerable groups [13,37,66,98]. The neglect of this preventive measure contributes to the maintenance of HBV circulation on a larger scale in these high-risk groups and may contribute to the emergence of new infections, co-infections, and mutations in the genome, such as those recorded in this study. Another important measure is the regular offer of diagnosis and treatment of PWUDs with HBV, including in remote areas in the Amazon region [13,37]. Despite the higher cost, in addition to serological tests to identify past exposure or ongoing HBV infection, there is a clear indication of the need to use PCR to screen for HBV due to the worrying rate of OBI observed in PWUDs in this Brazilian region, including PWUDs with OBI and vaccine escape mutations.

Finally, the strategies for health promotion to PWUDs provided for in the current Brazilian legislation must be fully applied in the Amazon region. In Brazil, multidisciplinary and outpatient psychosocial care centers for alcohol and drug users (Centro de Atenção Psicossocial Álcool e Drogas (CAPS-AD)) are available for people who use licit and illicit drugs in major Brazilian cities. In small cities, especially those located in remote areas, social and health services for PWUDs are very limited and often lacking. This public psychosocial and health care service is essential for (multi-) drug dependence/disorder, as well as for promoting the health of people who use psychotropic drugs, to reduce exposure and risks related to drug use, and adverse health consequences. Despite its limitations, engagement in drug treatment can reduce risky behaviors related to sex and drug use and, thus, reduce the acquisition and transmission of pathogens such as HBV, HCV, HDV, and HIV. Regular attendance of PWUDs at CAPS-AD can significantly facilitate the targeting of PWUDs for diagnosis, clinical monitoring, and treatment of people with HBV. Other strategies for health promotion have been suggested for PWUDs: (i) regular use of the Ambulatory Street strategy, and (ii) application of the HBV vaccine with flexible dosages through an accelerated schedule (0, 7, and 28 days) [37]. The implementation of these strategies is very important to reduce the spread of HBV and other viruses in the Brazilian Amazon, especially in high-risk populations such as PWUDs.

This study has limitations to be considered. Despite the authors describing the viral load values, only the qualitative results of the PCR assays for HBV and HDV-HBV were considered. PCR assays were conducted in the municipalities of Belém and Bragança. For this, there was a complex logistics for storing and transporting samples from remote areas of the Amazon region to the two municipalities mentioned, including with more five days of travel in different means of transport (small planes, small boats, and cars). The combination of high time for adequate processing of samples in the laboratories and the instability of the genetic material (e.g., RNA) may have impaired the viral load record and may also have caused false-negative results (lower than the detection limit of the PCR assay), mainly related to HDV-HBV co-infection. It should also be noted that this study screened for HDV only in samples with HBV DNA. HDV can suppress HBV replication [99] and therefore, failing to screen HBV DNA-negative samples, could result in an underestimation of the prevalence of HDV among PWUDs. Furthermore, this epidemiological report did not clinically evaluate the study subjects. It just assessed the presence of the viruses, advised, and directed all PWUDs infected with HBV and HDV for care in the public health network. Lastly, to date, numerous subtyping misclassifications have been documented for HBV, predominantly driven using partial genome sequences rather than full-length genomes [100].

In summary, the present study provides worrying information about the epidemiological status of HBV and HDV-HBV among people who use illicit drugs in the Amazon region, with implications for urgently improving HBV diagnosis, prevention, and treatment, and consequently reducing HBV-HDV co-infection.

## Figures and Tables

**Figure 1 viruses-13-01380-f001:**
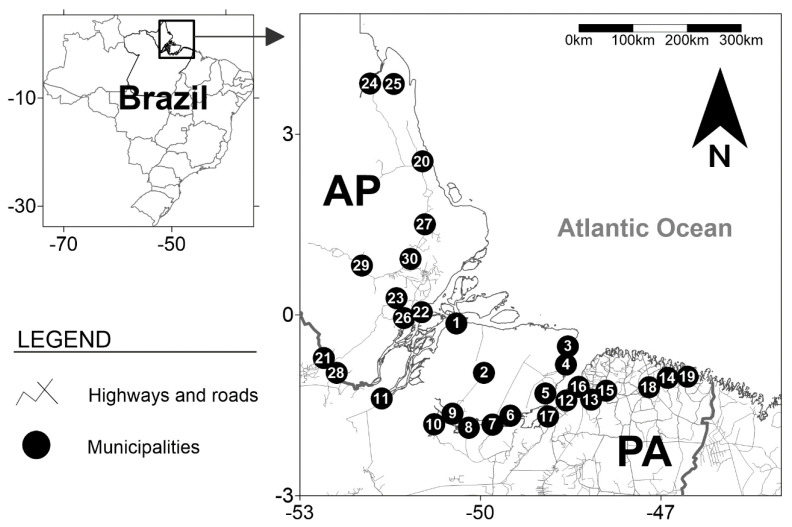
Geographical locations of the municipalities in the Amazon region where the people who used illicit drugs (PWUDs) were recruited for the study. The Brazilian states of Amapá and Pará are represented by the abbreviations AP and PA, respectively. The numbers from 1 to 30 indicate the municipalities where blood samples were collected, and information was provided by the PWUDs: (1) Afuá [number of PWUDs (*n*) = 21], (2) Anajás [*n* = 22], (3) Soure [*n* = 27], (4) Salvaterra [*n* = 23], (5) Ponta de Pedras [*n* = 18], (6) São Sebastião da Boa Vista [*n* = 19], (7) Curralinho [*n* = 85], (8) Bagre [*n* = 25], (9) Breves [*n* = 180], (10) Melgaço [*n* = 22], (11) Gurupá [*n* = 17], (12) Belém [*n* = 105], (13) Benevides [*n* = 21], (14) Bragança [*n* = 38], (15) Castanhal [*n* = 24], (16) Marituba [*n* = 59], (17) Abaetetuba [*n* = 20], (18) Capanema [*n* = 20], (19) Augusto Correa [*n* = 20], (20) Calçoene [*n* = 17], (21) Laranjal do Jari [*n* = 23], (22) Macapá [*n* = 69], (23) Mazagão [*n* = 28], (24) Oiapoque [*n* = 27], (25) Porto Grande [*n* = 18], (26) Santana [*n* = 55], (27) Tartarugalzinho [*n* = 15], (28) Vitória do Jari [*n* = 18], (29) Pedra Branca do Amapari [*n* = 17], and (30) Ferreira Gomes [*n* = 21].

**Table 1 viruses-13-01380-t001:** Epidemiological characteristics of the sample of PWUDs in the Amazon region.

Characteristics	N	%
Total sample	1074	100.0
Sex		
Male	704	65.6
Female	370	34.4
Age		
18–29 years	570	53.1
30–39 years	376	35.0
40+ years	128	11.9
Origin		
Born in the Amazon region	937	87.2
Not born in the Amazon region	137	12.8
Race/ethnicity (self-identified)		
White	355	33.1
Non-white (black + mixed race)	719	66.9
Sexual orientation		
Heterosexual	978	91.0
Same sex (including bisexual)	96	9.0
Marital status ^†^		
Single, separated, or widowed	678	63.1
Married or cohabitating	396	36.9
Education		
No formal education/some elementary school	622	57.9
Completed elementary school or higher	452	42.1
Monthly income ^†^		
Up to one Brazilian minimum wage	718	66.9
More than one Brazilian minimum wage	356	33.1
Main illicit drug used ^†^		
Marijuana	82	7.7
Crack/oxi	547	50.9
Cocaine	445	41.4
Daily use of illicit drugs ^†^	761	70.9
>5 years of illicit drugs use	303	28.2
Injecting drug use ^‡^	138	12.8
Sharing of drug use equipment ^†^	674	62.8
Detention (by police or in prison) ^†^	268	25.0
Unprotected sexual intercourse ^†^	882	82.1
More than 10 sexual partners ^†^	580	54.0
Sex work involvement ^†^	517	48.1
Blood transfusion history	165	15.4
Tattoos	667	62.1

^†^ Last 12 months. ^‡^ Last 24 months.

**Table 2 viruses-13-01380-t002:** Laboratory markers, genotypes, and subgenotypes of HBV among PWUDs.

Diagnosis	Number of Positive Cases/Total	%	95% CI
Serological markers	346/1074	32.2	29.8–34.9
HBsAg	18/1074	1.7	0.0–4.8
Anti-HBc	84/1074	7.8	5.8–10.1
HBsAg + Anti-HBc	32/1074	3.0	0.0–7.0
Anti-HBc + Anti-HBs	188/1074	17.5	15.6–19.6
Anti-HBs	24/1074	2.2	0.0–5.3
HBV DNA	77/1074	7.2	4.2–10.5
Genotype A	41/77	53.2	51.2–55.0
Subgenotype A1	35/77	45.4	43.5–48.0
Subgenotype A2	6/77	7.8	4.3–10.9
Genotype D	23/77	29.9	27.4–32.6
Subgenotype D2	4/77	5.2	1.3–8.1
Subgenotype D3	8/77	10.4	6.8–13.6
Subgenotype D4	11/77	14.3	10.2–18.1
Genotype F	13/77	16.9	13.5–20.6
Subgenotype F1a	3/77	3.9	0.0–7.4
Subgenotype F1b	1/77	1.3	0.0–4.1
Subgenotype F2a	6/77	7.8	4.3–10.9
Subgenotype F2b	1/77	1.3	0.0–4.1
Subgenotype F3	2/77	2.6	0.0–6.2

**Table 3 viruses-13-01380-t003:** Laboratory markers and genotypes of HDV among PWUDs.

Diagnosis	Number of Positive Cases/Total	%	95% CI
Anti-HDV antibodies	15/77	19.5	15.3–23.3
HDV RNA	9/77	11.7	7.5–15.5
Genotype 1	1/9	11.1	5.9–16.6
Genotype 3	8/9	88.9	86.1–92.2

**Table 4 viruses-13-01380-t004:** HBV subgenotypes identified among PWUDs with HDV RNA.

Identification	HBV Subgenotype	HDV Genotype
PWUD012AP	A2	3
PWUD014PA	A2	3
PWUD019AP	F2a	3
PWUD030PA	A1	3
PWUD035PA	F1a	3
PWUD054PA	A1	1
PWUD072PA	A1	3
PWUD073AP	A1	3
PWUD074PA	F1a	3

**Table 5 viruses-13-01380-t005:** Mutations identified in nucleotide sequences of HBV P and S genes in HBV-infected PWUDs.

Identification	HBV Subgenotype	Mutation (Relevance)	N/Total	%
PWUD037PA	F2b	G130R (escape vaccine)	1/77	1.3%
PWUD046AP	A1	P120L (escape vaccine)	1/77	1.3%
PWUD022PA	D2	A128V (escape vaccine)	2/77	2.6%
PWUD077PA				
PWUD070AP PWUD071AP PWUD073AP	A1	M204I (resistance to antivirals)	3/77	3.9%

## Data Availability

Epidemiological data are available from the corresponding author upon request. The nucleotide sequences obtained in this study were deposited in GenBank under access numbers MZ065397–MZ065473 (HBV) and MZ065474–MZ065482 (HDV).

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
