# Peer review of "Detection and Genetic Characterization of Hepatitis B and D Viruses: A Multi-Site Cross-Sectional Study of People Who Use Illicit Drugs in the Amazon Region"

_viruses, 2021, doi:10.3390/v13071380_

Round 1
Reviewer 1 Report
The paper entitled "Detection and genetic characterization of hepatitis B and D viruses: a multi-site cross-sectional study of people who use illicit drugs in the Amazon region " deals with the study of the presence of HBV and HBV HDV in the Brazilian Amazon region and the search for genotype/subgenotype qualification , mutation occurence and OBI qualification.The paper is well written and the technical approaches are sound. On the phylogenic tree for HBV the asignation of genotypes is OK but the topology of the tree is not clear. One may suggest to add an outgroup like the Woolly Monkey HBV to try to ameliorate the tree topology. Moreover the letters end * are very small and difficult to read. The approach classifying the genotypes with the 1300nt fragment is fine.
The tree for the HDV sequences is much better on a presentation aspect. The approach is rigourous and the tree informative. The HDV sequence lenght is sufficient for HDV classification.
On the discussion section, we would suggest the authors to discuss the repartition of HBV genotypes or sub-genotypes depending on the OBI status or not OBI status.
Otherwise an interesting question concerns the association between HBV and HDV genotypes and wether HDV genotype 3 is or not associated to a more severe clinical presentation especially in the presence of HBV genotype F?
Author Response
Dear Editor and Reviewers,
Please find enclosed the revised manuscript ID Viruses-1166321 entitled “Detection and genetic characterization of hepatitis B and D viruses: a multi-site cross-sectional study of people who use illicit drugs in the Amazon region”.
Most requests from the three reviewers were made. Those requests not made were duly justified. Then, each of the requests made by reviewers 1, 2 and 3 were commented point by point and sent by the platform to them. Some references were included and other excluded. The numbering of all references has been revised in the new version of the manuscript. Minor adjustments were also made to facilitate the presentation and understanding of the study. All changes were highlighted in blue in the revised manuscript. The authors thank the editor and reviewers for their attention and contribution to improving the presentation of the study.
Waiting for the decision of the editor and the reviewers.
Sincerely,
Prof. Dr. Aldemir Branco de Oliveira Filho.
Instituto de Estudos Costeiros, Universidade Federal do Pará.
Rua Leandro Ribeiro, s/n. Aldeia. Bragança PA, Brasil.
E-mail: olivfilho@ufpa.br. Telefone/Fax: + 55 91 3425 1209.
COMMENTS - REVIEWER 1:
“The paper entitled "Detection and genetic characterization of hepatitis B and D viruses: a multi-site cross-sectional study of people who use illicit drugs in the Amazon region " deals with the study of the presence of HBV and HBV HDV in the Brazilian Amazon region and the search for genotype/subgenotype qualification, mutation occurence and OBI qualification. The paper is well written and the technical approaches are sound. On the phylogenic tree for HBV the asignation of genotypes is OK but the topology of the tree is not clear. One may suggest to add an outgroup like the Woolly Monkey HBV to try to ameliorate the tree topology. Moreover the letters end * are very small and difficult to read. The approach classifying the genotypes with the 1300nt fragment is fine.
The tree for the HDV sequences is much better on a presentation aspect. The approach is rigourous and the tree informative. The HDV sequence lenght is sufficient for HDV classification.
On the discussion section, we would suggest the authors to discuss the repartition of HBV genotypes or sub-genotypes depending on the OBI status or not OBI status.
Otherwise an interesting question concerns the association between HBV and HDV genotypes and wether HDV genotype 3 is or not associated to a more severe clinical presentation especially in the presence of HBV genotype F?”
Responses to comments made by the reviewer 1:
1. “On the phylogenic tree for HBV the asignation of genotypes is OK but the topology of the tree is not clear. One may suggest to add an outgroup like the Woolly Monkey HBV to try to ameliorate the tree topology. Moreover the letters end * are very small and difficult to read”.
Reply to comment: The reviewer's suggestion was made. A representative outgroup sequence (AY226578 - Woolly monkey hepatitis B virus) was included in the alignment and topology of the HBV phylogenetic tree. Taxon name letters have been enlarged and the figure resolution has been increased as much as possible. Kindly see the new figure S1.
2. “On the discussion section, we would suggest the authors to discuss the repartition of HBV genotypes or sub-genotypes depending on the OBI status or not OBI status”.
Reply to comment: The authors included the distribution of HBV subgenotypes among PWUDs with OBI in the results. And, as suggested by the reviewer, this distribution was discussed in the manuscript. New texts added to manuscripts: “Among 27 PWUDs with OBI, subgenotypes A1 (55.6%), A2 (7.4%), D2 (3.7%), D3 (7.4%), D4 (18.5%), F1a (3.7%) and F1b (3.7%) were detected”, and “Among people with OBI, the presence of genotypes A (A1 and A2) and F (F2) has been recorded in the Amazon region [42]. However, this study detected a greater number of HBV subgenotypes, such as A1, A2, D2, D3, D4, F1a and F1b”.
3. “Otherwise an interesting question concerns the association between HBV and HDV genotypes and wether HDV genotype 3 is or not associated to a more severe clinical presentation especially in the presence of HBV genotype F?”.
Reply to comment: The authors cannot answer this question, as there was no clinical evaluation of the participants. However, we are grateful for the reviewer's comment, the manuscript has been modified to adequately state the facts and avoid such doubt. (“To date, there is little evidence of the implications of the association of genotypes A and 3 on the clinical course of co-infected individuals, only a report of a lower viral load [95]. Interestingly, genotypes F and 3 have already been associated with one of the serious forms of this co-infection, fulminant hepatitis [95-97]. In this study, three PWUDs with HBV-HDV were diagnosed with genotypes F and 3, but there was no clinical evaluation of these participants”, and “Furthermore, this epidemiological report did not clinically evaluate the study subjects. It just assessed the presence of the viruses, advised, and directed all PWUDs infected with HBV and HDV for care in the public health network”).
Reviewer 2 Report
This is a comprehensive study on the prevalence of HBV and HDV in the PWUD in the Amazon region with some interesting points. It is generally well-written. However, there are a few items that require clarification and/or confirmation. There are also some more minor English grammatical/phrasing issues outlined below.
- The authors only screened HBV DNA-positive samples for antibodies to HDV. HDV can suppress HBV replication and is often detected when HBV DNA is not detected (in up to 50% cases). This may result in a large underestimation of the true prevalence of HDV in this cohort. Is it possible to go back and screen all HBsAg-positive, HBV DNA-negative samples for HDV biomarkers? If not, then this over-sight needs to be acknowledged.
- The prevalence of OBI is quite high. Can the authors confirm that there was no contamination in the HBV PCR, resulting in inflated HBV DNA-positive, HBsAg-negative numbers? Were these samples tested twice with separate DNA extractions to confirm the first positive result? If so, is there an explanation for why these patients had no serological biomarkers such as immune suppression or HIV co-infection?
- The “vaccine escape” mutations in Table 5 are not the commonly described ones. The authors need to reference where they have sourced these. P120 is the site of a vaccine escape but the typical mutation is P120T not L, for example.
- Line 319: Suggest not including tenofovir here as drug-resistant mutations are not commonly described for it.
Minor Points:
- Line 54: “presents” is not quite the right word, ?use “possesses”
- Line 57: Instead of “present”, suggest writing “have several”
- Line 60: Define what S/S, P, C/C are?
- Lines 82, 177, 242: “analyzes” should be “analyses”;
- Line 89: “inadequate surgical…….” would read better as “inadequate screening before surgical procedures and blood transfusion in endemic areas”
- Line 100: Replace “displayed” with “found”
- Line 126: abbreviation for surface antigen was introduced on previous page
- Line 135: “anti-HDV” should be “anti-HD”
- Line 140: mention that this is an in-house PCR assay as previously described
- Lone 146: Replace “with” with “of”
- Lines 147, 237: “HDV RNA” more commonly used than “cDNA-HDV”
- Line 210: Replace :besides” with “In addition”
- Line 211: add “detectable” to HBV DNA
- Line 229: Remove “on the other hand”
- Line 249: Replace “They” with “This”
- Line 274: add screening “programs….do not use”
- Lines 276/7: Use either subject’s life history or subjects’ life histories
- Line 277: Suggest use “accurately” instead of “safely”
- Line 285: migration, not migrations
- Lines 288/9: suggest replacing “accelerated the proliferation” with “”contributed to the variety”
- Line 312: Replace “and around” with “of”
- Line 340: delete “Besides’
- Lines 371/2: Add HDV
- Line 395: use “PWUD”
Author Response
Dear Editor and Reviewers,
Please find enclosed the revised manuscript ID Viruses-1166321 entitled “Detection and genetic characterization of hepatitis B and D viruses: a multi-site cross-sectional study of people who use illicit drugs in the Amazon region”.
Most requests from the three reviewers were made. Those requests not made were duly justified. Then, each of the requests made by reviewers 1, 2 and 3 were commented point by point and sent by the platform to them. Some references were included and other excluded. The numbering of all references has been revised in the new version of the manuscript. Minor adjustments were also made to facilitate the presentation and understanding of the study. All changes were highlighted in blue in the revised manuscript. The authors thank the editor and reviewers for their attention and contribution to improving the presentation of the study.
Waiting for the decision of the editor and the reviewers.
Sincerely,
Prof. Dr. Aldemir Branco de Oliveira Filho.
Instituto de Estudos Costeiros, Universidade Federal do Pará.
Rua Leandro Ribeiro, s/n. Aldeia. Bragança PA, Brasil.
E-mail: olivfilho@ufpa.br. Telefone/Fax: + 55 91 3425 1209.
COMMENTS – REVIEWER 2:
“This is a comprehensive study on the prevalence of HBV and HDV in the PWUD in the Amazon region with some interesting points. It is generally well-written. However, there are a few items that require clarification and/or confirmation. There are also some more minor English grammatical/phrasing issues outlined below.
The authors only screened HBV DNA-positive samples for antibodies to HDV. HDV can suppress HBV replication and is often detected when HBV DNA is not detected (in up to 50% cases). This may result in a large underestimation of the true prevalence of HDV in this cohort. Is it possible to go back and screen all HBsAg-positive, HBV DNA-negative samples for HDV biomarkers? If not, then this over-sight needs to be acknowledged.
The prevalence of OBI is quite high. Can the authors confirm that there was no contamination in the HBV PCR, resulting in inflated HBV DNA-positive, HBsAg-negative numbers? Were these samples tested twice with separate DNA extractions to confirm the first positive result? If so, is there an explanation for why these patients had no serological biomarkers such as immune suppression or HIV co-infection?
The “vaccine escape” mutations in Table 5 are not the commonly described ones. The authors need to reference where they have sourced these. P120 is the site of a vaccine escape but the typical mutation is P120T not L, for example.
Line 319: Suggest not including tenofovir here as drug-resistant mutations are not commonly described for it”.
Minor Points:
Line 54: “presents” is not quite the right word, ?use “possesses”
Line 57: Instead of “present”, suggest writing “have several”
Line 60: Define what S/S, P, C/C are?
Lines 82, 177, 242: “analyzes” should be “analyses”;
Line 89: “inadequate surgical…….” would read better as “inadequate screening before surgical procedures and blood transfusion in endemic areas”.
Line 100: Replace “displayed” with “found”.
Line 126: abbreviation for surface antigen was introduced on previous page.
Line 135: “anti-HDV” should be “anti-HD”.
Line 140: mention that this is an in-house PCR assay as previously described.
Lone 146: Replace “with” with “of”
Lines 147, 237: “HDV RNA” more commonly used than “cDNA-HDV”
Line 210: Replace :besides” with “In addition”
Line 211: add “detectable” to HBV DNA
Line 229: Remove “on the other hand”.
Line 249: Replace “They” with “This”.
Line 274: add screening “programs….do not use”.
Lines 276/7: Use either subject’s life history or subjects’ life histories.
Line 277: Suggest use “accurately” instead of “safely”.
Line 285: migration, not migrations.
Lines 288/9: suggest replacing “accelerated the proliferation” with “”contributed to the variety”.
Line 312: Replace “and around” with “of”.
Line 340: delete “Besides’.
Lines 371/2: Add HDV.
Line 395: use “PWUD”.
Responses to comments made by the reviewer 2:
1. “The authors only screened HBV DNA-positive samples for antibodies to HDV. HDV can suppress HBV replication and is often detected when HBV DNA is not detected (in up to 50% cases). This may result in a large underestimation of the true prevalence of HDV in this cohort. Is it possible to go back and screen all HBsAg-positive, HBV DNA-negative samples for HDV biomarkers? If not, then this over-sight needs to be acknowledged”.
Reply to comment: At this time, we are unable to perform the assessment suggested by the reviewer (screen all HBV DNA negative samples). However, we recognize the importance of this information. Therefore, the authors included a new study limitation. New text in manuscript: “It should also be noted that this study screened for HDV only in samples with HBV DNA. HDV can suppress HBV replication [95] and this can result in an underestimation of the prevalence of HDV among PWUDs”.
2. “The prevalence of OBI is quite high. Can the authors confirm that there was no contamination in the HBV PCR, resulting in inflated HBV DNA-positive, HBsAg-negative numbers? Were these samples tested twice with separate DNA extractions to confirm the first positive result? If so, is there an explanation for why these patients had no serological biomarkers such as immune suppression or HIV co-infection?”.
Reply to comment: We appreciate your comments. All cases of exposure to HBV (including cases of OBI) and HBV-HDV have been tested twice to confirm the occurrence of the events. There was no disagreement between the first and second results. To avoid such doubts, this information was included in the manuscript (New text in manuscript: “All cases of exposure to HBV (including cases of OBI) and HBV-HDV were tested twice to confirm the occurrence of the events”). Unfortunately, we were unable to assess the presence of HIV in all samples in this study. On the other hand, the study with 466 PWUDs in the Marajó Archipelago, in the state of Pará (Reference 13 - https://doi.org/10.1007/s00705-016-3060-z), several factors were associated with HBV infection, among which the presence of HIV. Perhaps HBV-HIV co-infection is a possible explanation for the low presence of serological markers. However, we are unable to confirm this fact in this manuscript.
3. “The “vaccine escape” mutations in Table 5 are not the commonly described ones. The authors need to reference where they have sourced these. P120 is the site of a vaccine escape but the typical mutation is P120T not L, for example”.
Reply to comment: The authors modified the manuscript as suggested by the reviewer. New text in manuscript: “Vaccine escape mutations detected in the present study are not commonly described in their places of origin, as is the case of G130R, site G130. P120T is more commonly described at position P120 than P120L observed in our results [82]. Interestingly, another record was made in Rio de Janeiro, where P120L was identified in a person with OBI and genotype D2 [81]. A128V mutation, A128 site, has been reported in India and in cases of OBI and genotype D2 [83]”.
4. “Line 319: Suggest not including tenofovir here as drug-resistant mutations are not commonly described for it”.
Reply to comment: The authors modified the manuscript as suggested by the reviewer. Tenefovir was withdrawn from the sentence. New text in manuscript: “Prolonged use of nucleotide analogs, such as lamivudine, adefovir dipivoxil, and entecavir, can cause the emergence of drug-resistant strains”.
5. Minor Points:
Line 54: “presents” is not quite the right word, ?use “possesses”
Line 57: Instead of “present”, suggest writing “have several”
Line 60: Define what S/S, P, C/C are?
Lines 82, 177, 242: “analyzes” should be “analyses”;
Line 89: “inadequate surgical…….” would read better as “inadequate screening before surgical procedures and blood transfusion in endemic areas”.
Line 100: Replace “displayed” with “found”.
Line 126: abbreviation for surface antigen was introduced on previous page.
Line 135: “anti-HDV” should be “anti-HD”.
Line 140: mention that this is an in-house PCR assay as previously described.
Lone 146: Replace “with” with “of”
Lines 147, 237: “HDV RNA” more commonly used than “cDNA-HDV”
Line 210: Replace :besides” with “In addition”
Line 211: add “detectable” to HBV DNA
Line 229: Remove “on the other hand”.
Line 249: Replace “They” with “This”.
Line 274: add screening “programs….do not use”.
Lines 276/7: Use either subject’s life history or subjects’ life histories.
Line 277: Suggest use “accurately” instead of “safely”.
Line 285: migration, not migrations.
Lines 288/9: suggest replacing “accelerated the proliferation” with “”contributed to the variety”.
Line 312: Replace “and around” with “of”.
Line 340: delete “Besides’.
Lines 371/2: Add HDV.
Line 395: use “PWUD”.
Reply to comment: All minor points indicated by the reviewer were duly corrected, as recommended. Examples of new texts in the manuscript: “HBV belongs to the Hepadnaviridae family and possesses a partial double-strand DNA genome of approximately 3.2kb”, and “Most genotypes have several subgenotypes that may lead to different outcomes in chronicity after infection, disease progression, and responses to IFNα treatment”.
Reviewer 3 Report
The authors propose a very interesting epidemiological study for detection and molecular characterization of hepatitis B and D viruses in the northern part of Brazil from a large cohort of 1074 patients who use illicit drugs. The patient panel is well distributed over 2 states of this part of Brazil and includes 30 cities. The analyzed markers for HBV and HDV infections are complete and well described. Results are clear even if some mistakes are present and precisions need to be added. Finally the discussion is well documented and their conclusion brings a very important message such as the HBV screening with HBV-DNA molecular test especially for blood transfusion and patients with high risks in order to detect occult HBV infection.
These 2 viruses have been already largely studied in Brazil, but considering the high number of drug user patients, never studied, the results showed complement our knowledge of the HBV and HDV in this region of the world. This study could be published with some recommendations and corrections.
Although all parts of this study are well explained, there are some few errors and several questions and comments need to be clarified.
Major comments
- The authors showed a large table (Table 1) with epidemiological characteristics of the 1074 PWUDs but they did not made statistical analyses of them in function of the HBV and HDV profiles in order to determine any risk factors or socio-demographic factors for those 2 infections. It will be interesting to explore it and indicate the result.
- In supplementary data, the figure S2 representing the phylogenetic tree for HDV strains seems very unusual as HDV-3 genotype was never branched along with HDV-7 genotype and the other human HDV genotypes are organized with different manner comparing to all studies with specific HDV phylogenetic analyses (example: Karimzadeh H. et al J. Viral Hepat. 2019 / Le Gal F. et al Hepatology 2017). Perhaps it’s due to the outgroup made with the Avian HDV like agent (NC040845) or an analytic error?
- In table 2 the detail (number) of HBV serological markers for HBV-DNA positive patients might be added for an easier reading of those specific patients.
- Into results section especially “Laboratory diagnosis of viruses”, we do not have the information if some HBV-DNA and HDV-RNA positive patients were HBsAg negative? Because, HDV needs production of HBs proteins for its morphogenesis and propagation.
- The HDV loads found in this study seem very low for all patients. Does the method used and referenced (46) was validated for genotype HDV-3 samples as the genome of this genotype presents the highest divergence compared to the others?
Minor comments
- Page1 line 33: It will be more readable to add the term “co-infection” after HBV-HDV.
- Page2 line 83: The authors indicate that HDV-1 and -3 genotype can be found in Brazil, but in 2016, HDV-8 african strains was described especially in the northern part of this country (Santos MD et al Virus Resear 2016), probably from trade slave. I think this might be mentioned.
- Page2 line 86: Sexual transmission for HDV has never been clearly described or documented.
- Page5 line 216: The authors described 11 patients without any HBV serological marker. Was it possible to realize HBc-IgM research in order to detect HBV primo-infection?
- Page5 line 207: It’s indicated that 50/1074 (4.7%) of the patients were HBsAg positive whereas in table 2 it’s indicated 18/1074 (1.7%). What is the right value?
- Page8 line 253: In this discussion part, the authors cannot write that 32.2 % of the PWUDs were exposed to HBV infection as these 32.2% include 2.2% patients with only Anti-HBs (vaccinated patients).
- Page8 line 257: The reference to the Rio de Janeiro study cannot be mentioned as lower to the 30% prevalence of HBV markers found in the present study. In “Rio” study they found between 27.1% to 55.8% of HBV markers.
Author Response
Dear Editor and Reviewers,
Please find enclosed the revised manuscript ID Viruses-1166321 entitled “Detection and genetic characterization of hepatitis B and D viruses: a multi-site cross-sectional study of people who use illicit drugs in the Amazon region”.
Most requests from the three reviewers were made. Those requests not made were duly justified. Then, each of the requests made by reviewers 1, 2 and 3 were commented point by point and sent by the platform to them. Some references were included and other excluded. The numbering of all references has been revised in the new version of the manuscript. Minor adjustments were also made to facilitate the presentation and understanding of the study. All changes were highlighted in blue in the revised manuscript. The authors thank the editor and reviewers for their attention and contribution to improving the presentation of the study.
Waiting for the decision of the editor and the reviewers.
Sincerely,
Prof. Dr. Aldemir Branco de Oliveira Filho.
Instituto de Estudos Costeiros, Universidade Federal do Pará.
Rua Leandro Ribeiro, s/n. Aldeia. Bragança PA, Brasil.
E-mail: olivfilho@ufpa.br. Telefone/Fax: + 55 91 3425 1209.
COMMENTS - REVIEWER 3:
“The authors propose a very interesting epidemiological study for detection and molecular characterization of hepatitis B and D viruses in the northern part of Brazil from a large cohort of 1074 patients who use illicit drugs. The patient panel is well distributed over 2 states of this part of Brazil and includes 30 cities. The analyzed markers for HBV and HDV infections are complete and well described. Results are clear even if some mistakes are present and precisions need to be added. Finally the discussion is well documented and their conclusion brings a very important message such as the HBV screening with HBV-DNA molecular test especially for blood transfusion and patients with high risks in order to detect occult HBV infection.
These 2 viruses have been already largely studied in Brazil, but considering the high number of drug user patients, never studied, the results showed complement our knowledge of the HBV and HDV in this region of the world. This study could be published with some recommendations and corrections.
Although all parts of this study are well explained, there are some few errors and several questions and comments need to be clarified.
Major comments
The authors showed a large table (Table 1) with epidemiological characteristics of the 1074 PWUDs but they did not made statistical analyses of them in function of the HBV and HDV profiles in order to determine any risk factors or socio-demographic factors for those 2 infections. It will be interesting to explore it and indicate the result.
In table 2 the detail (number) of HBV serological markers for HBV-DNA positive patients might be added for an easier reading of those specific patients.
Into results section especially “Laboratory diagnosis of viruses”, we do not have the information if some HBV-DNA and HDV-RNA positive patients were HBsAg negative? Because, HDV needs production of HBs proteins for its morphogenesis and propagation.
The HDV loads found in this study seem very low for all patients. Does the method used and referenced (46) was validated for genotype HDV-3 samples as the genome of this genotype presents the highest divergence compared to the others?
Minor comments
Page1 line 33: It will be more readable to add the term “co-infection” after HBV-HDV.
Page2 line 83: The authors indicate that HDV-1 and -3 genotype can be found in Brazil, but in 2016, HDV-8 african strains was described especially in the northern part of this country (Santos MD et al Virus Resear 2016), probably from trade slave. I think this might be mentioned.
Page2 line 86: Sexual transmission for HDV has never been clearly described or documented.
Page5 line 216: The authors described 11 patients without any HBV serological marker. Was it possible to realize HBc-IgM research in order to detect HBV primo-infection?
Page5 line 207: It’s indicated that 50/1074 (4.7%) of the patients were HBsAg positive whereas in table 2 it’s indicated 18/1074 (1.7%). What is the right value?
Page8 line 253: In this discussion part, the authors cannot write that 32.2 % of the PWUDs were exposed to HBV infection as these 32.2% include 2.2% patients with only Anti-HBs (vaccinated patients).
Page8 line 257: The reference to the Rio de Janeiro study cannot be mentioned as lower to the 30% prevalence of HBV markers found in the present study. In “Rio” study they found between 27.1% to 55.8% of HBV markers”.
Responses to comments made by the reviewer 3:
1. “Major comments
The authors showed a large table (Table 1) with epidemiological characteristics of the 1074 PWUDs but they did not made statistical analyses of them in function of the HBV and HDV profiles in order to determine any risk factors or socio-demographic factors for those 2 infections. It will be interesting to explore it and indicate the result”.
Reply to comment: The authors understand the importance of the information requested by the reviewer. However, they cannot perform such an analysis and expose the findings related to HBV in this manuscript, as the factors associated with exposure to HBV have been published in other articles from the research group: PWUDs in the Marajó Archipelago (reference 13), PWUDs in the state of Amapá (reference 36), and non-injecting drug users in Pará (reference 38). As a result, the authors request that the manuscript focus on the prevalence of HBV and HDV infections, the distribution of genotypes of these viruses and the occurrence of relevant mutations.
2. “In table 2 the detail (number) of HBV serological markers for HBV-DNA positive patients might be added for an easier reading of those specific patients”.
Reply to comment: Table 2 was modified as suggested by the reviewer (see new table 2).
3. “Into results section especially “Laboratory diagnosis of viruses”, we do not have the information if some HBV-DNA and HDV-RNA positive patients were HBsAg negative? Because, HDV needs production of HBs proteins for its morphogenesis and propagation.
The HDV loads found in this study seem very low for all patients. Does the method used and referenced (46) was validated for genotype HDV-3 samples as the genome of this genotype presents the highest divergence compared to the others?”
Reply to comment: The authors included in the manuscript the information requested by the reviewer. New texts in manuscript: “In addition, 77 (7.2%) PWUDs had detectable HBV DNA. Viral load ranged from 5,6 x 102 to 3,1 x 104HBV copies/ml. All PWUDs with positive results for HBsAg (n = 50) also had positive results for HBV DNA (viral load: 1,2 x 104 and 3,1 x 104 HBV copies/ml) (Table 2). Among the 1024 PWUDs with negative results for HBsAg, five PWUDs with positive results only for the anti-HBc marker and 11 subjects with positive results for both anti-HBc and anti-HBs markers were also found to have HBV DNA (viral load: 5,6 x 102 to 4,8 x 103 HBV copies/ml)”; and “Among PWUDs with HBV DNA (50 with positive results for HBsAg, and 25 with negative results for HBsAg), 15 (19.5%) had anti-HDV antibodies and nine (11.7%) had HDV-RNA (viral load: 8.4 x 102 to 6.8 x 103 HDV copies/ml) (Table 4). All PWUDs with positive results for HDV also had tested positive for HBsAg”.
According to Kodani et al. (reference 46), the assay has the potential to detect all eight HDV genotypes. The authors believe in the sensitivity and specificity of the molecular assay for detection of HDV RNA. However, the authors suspect the occurrence of viral RNA degradation due to the long time taken to transport the samples from a very distant municipality to the laboratories located in the municipalities of Belém and Bragança (diagnosis of HDV RNA). Perhaps this could be related to the low HDV viral load detected in PWUDs with HDV RNA. This fact was exposed in the limitations of the study (“PCR assays were conducted in the municipalities of Belém and Bragança. For this, there was a complex logistics for storing and transporting samples from remote areas of the Amazon region to the two municipalities mentioned, including with more five days of travel in different means of transport (small planes, small boats, and cars). The combi-nation of high time for adequate processing of samples in the laboratories and the in-stability of the genetic material (RNA e.g.) may have impaired the viral load record and may also have caused false negative results (less than detection limit of the PCR assay), mainly related to HDV-HBV co-infection”).
4. Page1 line 33: It will be more readable to add the term “co-infection” after HBV-HDV.
Reply to comment: The change has been made (“The present study assessed the presence of HBV and HBV-HDV co-infection, identified viral sub-genotypes…”).
5. Page2 line 83: The authors indicate that HDV-1 and -3 genotype can be found in Brazil, but in 2016, HDV-8 african strains was described especially in the northern part of this country (Santos MD et al Virus Resear 2016), probably from trade slave. I think this might be mentioned.
Reply to comment: The change was made as recommended by the reviewer (“HDV Genotypes 1, 3 and 8 are found in South America, with genotype 3 considered to be original and prevalent in the Amazon region”).
6. Page2 line 86: Sexual transmission for HDV has never been clearly described or documented.
Reply to comment: The authors understand the reviewer's statement. However, there is evidence for sexual transmission, and people with high-risk sexual activity are at increased risk for infection as can be seen in the references: Hughes et al. 2011 - Reference 27; Wu et al. 1995 - https://doi.org/10.1002/hep.1840220607; Rosenblum et al. 1992 – JAMA 1992, 267(18):2477-2481; Chen et al. 2018 - http://dx.doi.org/10.1136/gutjnl-2018-316601; Vlachogiannakos & Papatheodoridis 2020 - https://doi.org/10.1111/liv.14357. Thus, the authors modified the manuscript, clearly showing the scientific evidence currently available. New text in manuscript: “HBV is transmitted by the sexual and parenteral routes, and HDV is transmitted by the parenteral route and with evidence of sexual transmission. People with high-risk sexual activity are at increased risk of HBV and HDV infection”.
7. Page5 line 216: The authors described 11 patients without any HBV serological marker. Was it possible to realize HBc-IgM research in order to detect HBV primo-infection?
Reply to comment: Total Anti-HBc determines the presence of both IgM and IgG class antibodies. In this study, the Murex anti-HBc kit (Total) (DiaSorin: sensitivity: 100.0%, and specificity: 99.7%) was used to detect HBc-IgG+ HBc-IgM antibodies. In the study it was not possible to test only for HBc-IgM. However, the authors highlight that all study samples were tested for the presence of HBV DNA, another important marker of HBV infection.
8. Page5 line 207: It’s indicated that 50/1074 (4.7%) of the patients were HBsAg positive whereas in table 2 it’s indicated 18/1074 (1.7%). What is the right value?
Reply to comment: Among the various information contained in Table 2, it indicates that 18 PWUDs (1.7%) tested positive for HBsAg (only) and 32 PWUDs (3.0%) tested positive for HBsAg + Anti-HBc. In total, the study identified 50 PWUDs (4.7%) with positive results for HBsAg.
9. Page8 line 253: In this discussion part, the authors cannot write that 32.2 % of the PWUDs were exposed to HBV infection as these 32.2% include 2.2% patients with only Anti-HBs (vaccinated patients).
Reply to comment: The authors are grateful for the correct reading of Table 2 and for indicating the error. The authors corrected the text as recommended by the reviewer (“The first important finding was that 30.0% of PWUDs were exposed to HBV, as indicated by the presence of either HBsAg or anti-HBc or both”).
10. Page8 line 257: The reference to the Rio de Janeiro study cannot be mentioned as lower to the 30% prevalence of HBV markers found in the present study. In “Rio” study they found between 27.1% to 55.8% of HBV markers.
Reply to comment: The authors are grateful for identifying the problem and the manuscript has been altered to resolve the disagreement (“This percentage is higher than the values reported for most PWUDs from two other Brazilian regions, such as Salvador (0.0 - 6.2%) in the Northeast, and Campo Grande, Cuiabá, and Goiânia (7% - 14.0%) in the Midwest”).
Round 2
Reviewer 2 Report
I am satisfied that the authors have addressed all my concerns/comments. I have 6 additional minor grammatical changes, see below. Line 57: change sequences to sequence Line 90: remove “since” Line 151: change “with” to “of’ Line 254: change “cDNA-HDV” to “HDV RNA” Lines 421-422: This would read better as: ”HDV can suppress HBV replication (ref) and therefore, failing to screen HBV DNA-negative samples, could result in an underestimation…..” Lines 337-338: Delete “in their places of origin,”